# RL FROM PHYSICAL FEEDBACK: ALIGNING LARGE MOTION MODELS WITH HUMANOID CONTROL

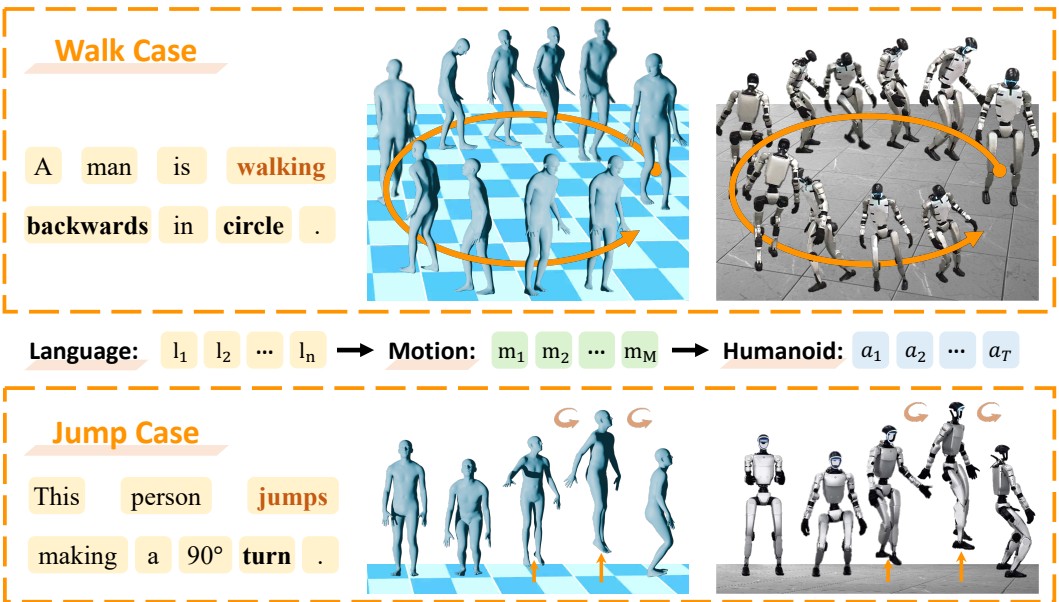

Figure 1: Two cases of RLPF. We establish a unified text–motion–humanoid pipeline, where the generated motions not only preserve strong text alignment but also substantially improve physical feasibility in both simulators and real-world deployments. Here, $l_i$ denotes the i-th language token, $m_i$ the i-th motion token produced by the large motion model, and $a_i$ the i-th action executed on the humanoid robot.

## ABSTRACT

This paper focuses on a critical challenge in robotics: translating text-driven generated human motions into executable actions for humanoid robots. While existing text-to-motion generation methods achieve semantic alignment between language and motion, they often produce kinematically or physically infeasible motions unsuitable for real-world deployment. To bridge the gap between motion generation and humanoid execution, we propose **Reinforcement Learning from Physical Feedback (RLPF)**, a novel framework that integrates physics-aware motion evaluation with text-conditioned motion generation. RLPF employs a motion tracking policy to assess feasibility in a physics simulator, generating rewards for fine-tuning the motion generator. Furthermore, RLPF introduces an alignment verification module to preserve semantic alignment to text instructions. This joint optimization ensures both physical feasibility and instruction alignment. Extensive experiments show that RLPF greatly outperforms baseline methods in generating physically feasible motions while maintaining semantic alignment with text instruction, enabling successful deployment on real humanoid robots. More visualizations are available at https://anonymous.4open.science/r/RLPF/.

# 1 INTRODUCTION

Humanoid robots have advanced rapidly in recent years, driven by improvements in actuators, torque control, and whole-body control policies. They are now capable of performing expressive whole-body behaviors with greater robustness and fluidity, when provided with reference trajectories (He et al., 2025; Truong et al., 2025; Wang et al., 2025). While such advances endow robots with powerful physical capabilities, this is only the physical side of the entire humanoid system. To achieve the intelligent side, a humanoid robot also requires another critical component—an intelligent "brain" that can translate open-form instructions into executable motions, serving as the bridge between high-level goals and a low-level controller.

One promising approach toward this goal is text-to-motion (T2M) generation (Tevet et al., 2022b; Yuan et al., 2023; Tevet et al., 2022a). By learning from human motion datasets with text labels, T2M models translate natural language into temporal sequences of human joint positions and rotations, which can, in principle, be retargeted to humanoid embodiments and then serve as reference trajectories. Previous works mainly relied on small-scale mocap datasets, such as AMASS (Mahmood et al., 2019) and HumanML3d (Guo et al., 2022). More recently, as larger motion datasets from Internet videos have emerged (Lin et al., 2023), large language models (LLMs) have been integrated with motion generation, evolving T2M models into large motion models (Wang et al., 2024b;a; Zhang et al., 2024b). These advances achieve greater motion diversity, improved generalization, and closer alignment with human intent.

However, when we apply T2M models as humanoid brains and integrate them with humanoid whole-body control policies, a crucial gap remains. Existing T2M research has primarily focused on semantic alignment with text descriptions and visual motion fidelity, while largely neglecting physical feasibility. As a result, generated motions often suffer from unrealistic artifacts such as skating, floating, and abrupt transitions, which severely limit their usability for humanoid control. Moreover, the reliance on Internet video–derived motion data further exacerbates this issue due to reconstruction errors introduced during video-to-motion conversion. Together, these limitations make current T2M models less ideal to serve as reliable humanoid brains capable of producing executable motions in real-world settings. Therefore, this paper attempts to address this core research question:

***How can we align motion generation models with whole-body control systems for humanoid robots, ensuring that generated motions maintain both semantic correspondence and physical feasibility?***

To address this challenge, we consider to incorporate physical constraints and feedback into the T2M training process, which could be essential for improving the physical feasibility of generated motion. Previous attempts have considered physical feedback in terms of metrics such as skating, floating, and penetration (Yuan et al., 2023; Han et al., 2025; Tseng et al., 2023). However, these metrics only provide indirect measures of physical feasibility and therefore cannot directly assess the quality of generated motion when deployed on humanoid robots. Instead, we propose to obtain physical feedback from motion tracking policies, in which the generated motions serve as reference inputs to the tracking policy, and the execution result in simulation offers a more direct and comprehensive evaluation of physical feasibility.

Building upon the assessment of physical feasibility, we further leverage this physical feedback to refine the motion generation model. Inspired by recent advances in reinforcement learning from human feedback (RLHF) (Ouyang et al., 2022; Shao et al., 2024), we regard simulator-derived tracking feedback as a reward signal and fine-tune pre-trained large motion models using reinforcement learning. However, our experiments show that optimizing solely for physical feasibility tends to produce overly conservative motions and leads to a catastrophic loss of semantic alignment with text instructions. For example, as illustrated in Figure 3, regardless of the input text, the model degenerates to generating only standing motions. To address this problem, we further introduce an Alignment Verification Module that explicitly preserves semantic correspondence with text.

More specifically, our approach consists of three key components: ***(i) Motion Tracking Policy***, which provides physical feasibility feedback in simulators; ***(ii) Alignment Verification Module***, which evaluates the semantic alignment between generated motions and textual instructions; and ***(iii) RL fine-tuning framework***, which optimizes the large motion model using both physical feasibility signals from the tracking policy and semantic alignment feedback from the verification module.

In summary, our key contributions are threefold:

- We propose **Reinforcement Learning from Physical Feedback (RLPF)**, a novel framework that optimizes large motion models through physical feedback from a motion tracking policy, significantly improving the physical feasibility of generated motions on humanoid robots.

- We introduce an alignment verification module that quantitatively assesses text-motion correspondence, enabling RLPF to maintain semantic alignment throughout the RLPF process.

- Through extensive experiments on AMASS and MotionX, we demonstrate that our approach achieves **state-of-the-art** physical feasibility while preserving strong semantic alignment, effectively bridging text-to-motion generation with real-world humanoid control.

## 2 RELATED WORKS

**Human Motion Generation.** The rapid development of this task is driven by the availability of high-quality, inscreasing-scale datasets with diverse motions and rich text descriptions. For instance, AMASS (Mahmood et al., 2019) unifies 15 motion capture datasets under a standardized framework. Building on this, HumanML3D (Guo et al., 2022) offers 14,616 motion sequences and 44,970 text descriptions spanning daily activities, sports, and artistic performances (totaling 28.59 hours). Further expanding this scale, MotionX (Lin et al., 2023) introduces 15.6M 3D pose annotations and 81K sequences, leveraging an automated annotation pipeline for efficient and precise data collection. Additionally, MOVI (Ghorbani et al., 2021) combines video recordings with real pose and shape data, supporting both generative and discriminative tasks. These datasets have facilitated the development of advanced text-to-motion models. T2M-GPT (Zhang et al., 2023) combines VQ-VAE (Van Den Oord et al., 2017) and generative transformers (GPT) to improve alignment with human intent. MotionGPT (Jiang et al., 2023) treats human motion as a "foreign language" by employing an LLM for unified motion-language modeling. Its successor, MotionGPT-2 (Wang et al., 2024b) extends this framework with multimodal control (e.g., text and pose inputs). Meanwhile, diffusion-based approaches have achieved high-quality motion synthesis. MotionDiffuse (Zhang et al., 2024a) generates diverse and realistic motions through iterative denoising, while the Motion Diffusion Model (MDM) (Tevet et al., 2022b) incorporates geometric losses (e.g., foot contact loss) from the motion generation domain to achieve state-of-the-art performance. Our approach leverages these advancements by utilizing pre-trained models to generate initial motion sequences, then fine-tuning their weights via reinforcement learning to produce physically feasible motions for robotics. Additionally, we optimize motion retargeting by minimizing the distance between human and robot corresponding points while preserving natural movement through regularization.

**Reinforcement Learning Fine-Tuning for Large Models.** RL has emerged as a powerful tool for fine-tuning large pre-trained models, optimizing their performance for specific tasks or aligning them with user preferences. In natural language processing, reinforcement learning from human feedback (RLHF) leverages human evaluation data — such as text summarization of tasks (Stiennon et al., 2020) — to train reward models and refine outputs using RL algorithms like Proximal Policy Optimization (PPO) (Schulman et al., 2017). This approach overcomes limitations of traditional supervised fine-tuning for complex objectives. In human motion generation, RL is increasingly adopted to enhance motion quality, generalization, and real-world applicability. For example, InstructMotion (Mao et al., 2024) frames text-to-motion generation as a Markov decision process, using contrastive pre-trained encoders (text and motion) to design rewards that improve generalization to novel descriptions and reduce overfitting caused by limited training data. Similarly, RL has been applied to motion control, such as training physics-based tracking policies via PPO for simulated environments (Luo et al., 2023). Inspired by these advances, we employs RL to fine-tune motion generation models, prioritizing physical feasibility through physical feedback. By leveraging pre-trained motion tracking policies to generate rewards, we optimize motions to respect robotic kinematic and dynamic constrains, bridging the gap between generative motion models and practical robotic execution.

## 3 RL FROM PHYSICAL FEEDBACK (RLPF)

This section outlines key components of our RLPF framework. First, we lay the foundation for RLPF by pre-training a large motion model (Section 3.1). Second, we introduce two essential reward components for RLPF: *i) a novel motion tracking reward* (Section 3.2), and *ii) an alignment verification module* to ensure the generation of physically feasible and semantically-aligned motions

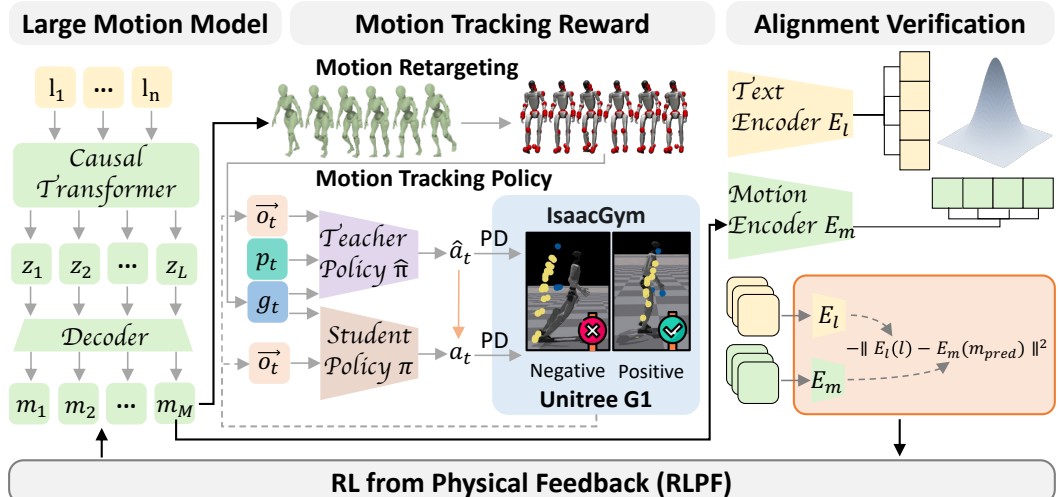

Figure 2: **Overview of RLPF**, which consists of three key components: *i) Large Motion Model* which is pre-trained on motion datasets as a foundation of RL fine-tuning;*ii) Motion Tracking Reward* which is pre-trained to evaluate the physical feasibility of generated motions; *iii) Alignment Verification Module* which enhances text-motion semantic consistency while preserving physical feasibility.

(Section 3.3). Finally, we present the core component of our framework: reinforcement learning (RL) fine-tuning of the pre-trained large motion model (Section 3.4).

## 3.1 PRE-TRAINING OF LARGE MOTION MODEL

**Motion Tokenizer.** Existing motion models (Jiang et al., 2023; Wang et al., 2024a) in this field typically employ vector quantization (VQ) (Van Den Oord et al., 2017) or its variants (Guo et al., 2024) to represent human motions, treating them as a "foreign language". These approaches use a motion tokenizer to discretize motion sequences into tokens. The tokenizer comprises three components: *i)* an encoder $\mathcal{E}$ that maps motion sequences to latent codes, *ii)* a decoder $\mathcal{D}$ that reconstructs motions from tokens, and *iii)* a codebook $\mathcal{C}$ where each codeword represents a distinct human body movement pattern. More specifically, Given a motion sequence with $M$ frames, the tokenizer first encodes it into motion tokens $\vec{z}_{1:L} = \{z_i\}_{i=1}^L$, then decodes them back to the motion $\vec{m}_{1:M} = \mathcal{D}(\vec{z}_{1:L}) = \mathcal{D}(\mathcal{E}(\vec{m}_{1:M}))$.

**Large Motion Model for T2M Generation.** Recent works have shown increasing interest in leveraging large language models (LLMs) for text-to-motion (T2M) generation, capitalizing on their ability to interpret human intent (Jiang et al., 2023). While our approach follows this paradigm, we emphasize that our framework is not restricted to any specific T2M methodology. In this work, our model extends the LLM's vocabulary with $K$ discrete codes from a motion codebook. Our LLM backbone $P_\Theta$, implemented as a decoder-only causal transformer, first encodes the input text tokens $\mathcal{L}$, then auto-regressively generates corresponding motion tokens $\vec{z}_{1:L} = \{z_i\}_{i=1}^L$. The entire training process of our model consists of two stages: i) large-scale pre-training on text-motion pairs, followed by ii) RL post-training. During pre-training, we optimize the model using negative log-likelihood minimization over predicted tokens:

$$\mathcal{L}(\Theta) = -\sum_{j=1}^{T} \log P_\Theta(z_j | l, \vec{z}_{1:j-1}) \tag{1}$$

where $l$ represents the text instruction, $\vec{z}_{1:j-1}$ and $z_j$ denote the first $j-1$ motion tokens and the $j$-th motion token to predict, respectively. $\Theta$ represents the model parameters of $P_\Theta$ and $T$ is the length of the target sequence.

## 3.2 MOTION TRACKING REWARD

After pre-training the large motion model, we introduce a motion tracking reward to provide a straightforward evaluation of how physically feasible the generated motions are, which facilitates subsequent RL fine-tuning. To compute this reward, we implement a three-stage pipeline: *i)* motion retargeting, *ii)* motion tracking and *iii)* offline evaluation.

### 3.2.1 MOTION RETARGETING

Due to the morphological differences between humans and humanoid robots, human motion must be adjusted before being used as goals for the tracking policy. Following the approach of H2O (He et al., 2024), we employ an optimization-based method for motion retargeting. Specifically, we first optimize the body shape parameters $\beta$ of the SMPL model (Loper et al., 2023) using gradient descent to align the SMPL geometric structure with that of the robot, thus obtaining a human body shape corresponding to the robot. With the body shape fixed, we then optimize the pose parameters $\theta$ and root translations iteratively by means of gradient descent. The objective is to minimize a loss function defined as the discrepancy between humanoid joint positions—computed via forward kinematics from the target pose and root translation—and the corresponding SMPL body joint positions.

### 3.2.2 MOTION TRACKING POLICY

After retargeting, the motion is morphologically consistent with the humanoid robot. Nevertheless, verifying its physical feasibility and transferability to the real world still requires a well-trained motion-tracking policy to evaluate the the tracking performance of this motion in simulation. To train such a tracking policy, following BumbleBee (Wang et al., 2025), we partition the motion dataset into multiple clusters. For each cluster, we train the tracking policy via a two-stage teacher–student framework. In the first stage, we learn an oracle teacher using PPO (Schulman et al., 2017) that leverages privileged, simulator-only information. In the second stage, we distill the teacher into a deployable student policy by removing privileged signals that are inaccessible in the real world. All training is performed in IsaacGym (Makoviychuk et al., 2021), enabling high-throughput parallel simulation. The details of our policy training procedure are as follows:

**Stage 1: Teacher Policy Training.** The teacher policy $\hat{\pi}$ is trained as an oracle in the simulator using PPO. It leverages privileged information $p_t$—unavailable in the real world—together with a history of observations $\vec{o}_t = o_{t-H:t}$ and motion-tracking targets $g_t$, which is the retargeted reference motion. The policy outputs target joint positions $\hat{a}_t \in \mathbb{R}^{23}$, which are executed by Proportional–Derivative (PD) controllers to produce humanoid actions.

**Stage 2: Student Policy Training.** The student policy $\pi$ distills the teacher's knowledge while receiving solely on real-world-observable inputs: a history of observations $\vec{o}_t = o_{t-H:t}$ and motion target $g_t$. Following a DAgger-like (Ross et al., 2011) approach, we first roll out the student in the simulator to collect both history of observations $\vec{o}_t$ and privileged information $p_t$. Then we ask the teacher $\hat{\pi}$ to compute oracle actions $\hat{a}_t$ at these states, serving as the supervision signal. The student is refined by minimizing an MSE loss $l = \|a_t - \hat{a}_t\|^2$ on collected data. The process iterates until convergence, producing a deployable policy without privileged information.

### 3.2.3 OFFLINE EVALUATION

After obtaining a deployable student policy, we conduct offline evaluation in the simulator (e.g., Isaac-Gym). The physical feasibility of generated motion is quantified using tracking success rates, which are then used as motion tracking rewards to optimize the large motion model through reinforcement learning. This process is formalized as:

$$R_{tracking}^{m_{pred}} = \mathbb{I}\left(Succ(\pi, m_{pred})\right) \tag{2}$$

where $\pi$ denotes the student policy as introduced in Section 3.2.2, $m_{pred}$ represents the generated motion, and $Succ(\pi, m_{pred})$ is a binary success flag to indicate whether the generated motion $m_pred$ is successfully executed by the policy $\pi$, where the failure conditions are defined as positional deviation exceeding threshold $\epsilon$ or loss of body balance.

### 3.3 MOTION ALIGNMENT VERIFICATION

We perform RL optimization using the motion tracking reward described in Section 3.2, aiming to generate motion sequences that are more physically feasible. However, as latter shown in the our experiment (Section 4.4), this single-objective optimization may compromise the semantic alignment with textual instructions. To maintain both semantic alignment and physical feasibility, we propose an Alignment Verification Module to regularize the large motion model using contrastive-based pretrained encoders ($E_m$ for motion and $E_l$ for text). Both encoders are trained using the following objective:

$$\mathcal{L}_{CL} = (1 - y)(\|f_l - f_m\|)^2 + y(max(0, \alpha - \|f_l - f_m\|))^2 \tag{3}$$

where $y$ is a binary label indicating text-motion pair matching, $\alpha$ is a margin threshold, and $f_l = E_l(l)$, $f_m = E_m(m)$ denote text and motion embeddings respectively. The encoders project both modalities into a shared space where semantic similarity corresponds to smaller Euclidean distances. We define the alignment verification as

$$R_{align}^{m_{pred}} = (\|E_l(l) - E_m(m_{pred})\|)^2 \tag{4}$$

where $R_{align}^{m_{pred}}$ denotes the alignment verification score, and $m_{pred}$ denotes the generated motion.

### 3.4 FINE-TUNING OF MOTION GENERATOR VIA RL

After obtaining the pretrained large motion model and the two reward functions, we employ reinforcement learning (RL) to fine-tune the model, thereby improving the physical feasibility while maintain and the textual alignment of the generated motions. We treat the motion generator $P_\Theta$ as the policy model and optimize it via Group Relative Policy Optimization (GRPO) (Shao et al., 2024). This RL algorithm improves upon PPO by removing the need for a critic model and estimating baselines directly from relative group scores. For each text instruction $l$ sampled from dataset distribution $P(L)$, GRPO operates by: i) sampling a group of motion token sequences $Z = \{\vec{z}^1, \vec{z}^2, \ldots, \vec{z}^G\}$ from the previous policy $P_{\Theta_{old}}$; ii) optimizing the current policy $P_\Theta$ by maximizing the following objective:

$$\mathcal{J}_{GRPO}(\theta) = \mathbb{E}[l \sim P(L), \{\vec{z}^i\}_{i=1}^G \sim P_{\Theta_{old}}(Z|l)]$$

$$\frac{1}{G}\sum_{i=1}^G \left( \min\left( \frac{P_\Theta(\vec{z}^i|l)}{P_{\Theta_{old}}(\vec{z}^i|l)} A_i, \text{clip}\left( \frac{P_\Theta(\vec{z}^i|l)}{P_{\Theta_{old}}(\vec{z}^i|l)}, 1 - \epsilon, 1 + \epsilon \right) A_i \right) - \beta \mathbb{D}_{KL}\left( P_\Theta || P_{\Theta_{ref}} \right) \right) \tag{5}$$

$$\mathbb{D}_{KL}\left( P_\Theta || P_{\Theta_{ref}} \right) = \frac{P_{\Theta_{ref}}(\vec{m}^i|l)}{P_\Theta(\vec{m}^i|l)} - \log \frac{P_{\Theta_{ref}}(\vec{m}^i|l)}{P_\Theta(\vec{m}^i|l)} - 1 \tag{6}$$

The optimization involves two key hyperparameters: $\epsilon$ controlling the policy update constraint, and $\beta$ balancing the KL-divergence penalty. Here, $P_{\Theta_{ref}}$ denotes the reference policy, which is the pre-trained large motion model, while $A_i$ represents the advantage computed from a group of rewards $\{r_1, r_2, \ldots, r_G\}$ across all generated outputs within the group:

$$A_i = \frac{r_i - \text{mean}(\{r_1, r_2, \cdots, r_G\})}{\text{std}(\{r_1, r_2, \cdots, r_G\})} \tag{7}$$

## 4 EXPERIMENTS

In this section, we conduct a comprehensive evaluation of our proposed method, organized as follows: First, we describe the experimental setup and baseline comparisons. We then analyze three key research questions:

- **Q1 (Feasibility Improvement)**: Does RLPF enable the large motion model to produce more physical feasible motions than supervised fine-tuning (SFT), even when SFT also uses physical feedback?
- **Q2 (Reward Analysis)**: How critical is the motion tracking reward to physical feasibility, and what are the performance consequences of modifying or removing this reward mechanism?
- **Q3 (Alignment Verification)**: Does our alignment verification effectively preserve text-motion semantic alignment?

## 4.1 EXPERIMENTAL SETUP

**Compared Baseline.** To the best of our knowledge, RLPF is the first attempt that employs reinforcement learning to optimize the motion generator for enhanced feasibility. We evaluate RLPF against three baseline approaches: *i) Base Model* which refers to the pretrained motion generator used as our foundation for fine-tuning. *ii) RobotMDM* (Serifi et al., 2024) which learns a critic model for physical feasibility and integrate it with SFT loss. To ensure a fair comparison, we transfer the RobotMDM method onto our the same LLM backbone. *iii) SFT-Filter* which refers to supervised fine-tuning only on tracking-optimized data, filtered by a motion tracking policy to retain only successfully trackable samples.

**Simulator Setting.** Following previous works on motion tracking policies (Ji et al., 2024; He et al., 2025), we perform a three-stage evaluation protocol. First, we conduct large-scale reinforcement learning training in IsaacGym. Second, we perform a zero-shot transfer to the MuJoCo (Todorov et al., 2012) simulator to assess the policy's cross-simulator generalization. Finally, we carry out physical deployment on a Unitree G1 humanoid robot to verify the real-world effectiveness of RLPF on motion tracking.

**Dataset.** To comprehensively validate the effectiveness of our proposed RLPF, we conduct experiments on the MoCap dataset AMASS and on MotionX-W, a walking-motion subset of the Internet video-based dataset MotionX. Following ASAP and BumbleBee, we apply PHC (Luo et al., 2023) to filter the dataset, yielding 8,179 motions from AMASS and 12,052 walking motions from the MotionX-W. However, it is worth noting that since PHC considers only limited physical constraints, the filtered motions are not guaranteed to be physically feasible for humanoid robots. We use LLaMA2-7B (Touvron et al., 2023) as our base large language model.

**Tracking Policies.** For the AMASS dataset, we train the tracking policy following BumbleBee. For the MotionX-W dataset, we fine-tune the policy pre-trained on AMASS to better adapt to MotionX-W data.

**Evaluation Metrics.** We consider both high-level motion generation and low-level motion tracking metrics to evaluate our motion generator. evaluate motion generators using both high-level motion generation metrics and low-level motion tracking metrics.

- **High-Level Motion Tracking Metrics.** These metrics evaluate text-to-motion alignment and the fidelity of generated motions compared to their ground truth. Following Guo et al. (2022), we consider text-motion retrieval metrics (R@1, R@3), Multimodal Distance (MMDist), Fréchet Inception Distance (FID) and diversity (DIV).
- **Low-Level Motion Generation Metrics.** We evaluate motions in IsaacGym and MuJoCo before real-robot deployment, aligning with OmniH2O (Ji et al., 2024) and PHC (Luo et al., 2023). Key metrics include: *i) Success Rate* ($Succ$), which evaluates whether the humanoid robot follows the reference motion without falling. A failure is recorded if the maximum deviation exceeds $0.5$, m at any timestep or if the robot is detected as fallen based on roll/pitch thresholds. *ii) Mean Per Joint Position Error* (MPJPE, $E_{mpjpe}$(rad)), which quantifies measures joint tracking accuracy. *iii) Mean Per Keybody Position Error* (MPKPE, $E_{mpkpe}$ (m)), which evaluates keypoint positional accuracy. Notably, $Succ$ is the primary metric, as it depends on both MPJPE and MPKPE and incorporates the body's pitch angle.

## 4.2 LOW-LEVEL TRACKING EVALUATION

To answer **Q1 (Feasibility Improvement)**, we first evaluate RLPF's performance on both the AMASS and MotionX-W test sets. As shown in Table 1 and Table 2, RLPF consistently outperforms all

Table 1: Comparisons of low-level tracking evaluation on the AMASS test set.

| Method | IsaacGym | | | MuJoCo | | |
|---|---|---|---|---|---|---|
| | $Succ\uparrow$ | $E_{mpjpe}\downarrow$ | $E_{mpkpe}\downarrow$ | $Succ\uparrow$ | $E_{mpjpe}\downarrow$ | $E_{mpkpe}\downarrow$ |
| **Base Model** | 0.57 | 0.24 | 0.15 | 0.53 | 0.30 | 0.34 |
| **SFT-Filter** | 0.60 | 0.25 | 0.14 | 0.55 | 0.28 | 0.32 |
| **RobotMDM** | 0.62 | 0.24 | 0.14 | 0.55 | 0.27 | 0.31 |
| **RLPF** | 0.83 | 0.21 | 0.08 | 0.78 | 0.25 | 0.26 |

Table 2: Comparisons of low-Level tracking evaluation on the MotionX-W test set.

| Method | IsaacGym | | | MuJoCo | | |
|---|---|---|---|---|---|---|
| | $Succ\uparrow$ | $E_{mpjpe}\downarrow$ | $E_{mpkpe}\downarrow$ | $Succ\uparrow$ | $E_{mpjpe}\downarrow$ | $E_{mpkpe}\downarrow$ |
| **Base Model** | 0.56 | 0.27 | 0.14 | 0.49 | 0.40 | 0.38 |
| **SFT-Filter** | 0.57 | 0.28 | 0.14 | 0.51 | 0.38 | 0.32 |
| **RobotMDM** | 0.62 | 0.28 | 0.13 | 0.58 | 0.39 | 0.28 |
| **RLPF** | 0.75 | 0.26 | 0.11 | 0.71 | 0.35 | 0.19 |

baseline methods across every evaluation metric in both datasets and simulators, clearly establishing its effectiveness.

## 4.3 HIGH-LEVEL GENERATION EVALUATION

To validate that our approach maintains high-quality motion generation, we evaluate it using high-level generation metrics. As shown in Table 3, motions generated by RLPF consistently exhibit robust text alignment and diversity. The experimental results demonstrate that our approach achieves significant improvements in low-level tracking while preserving high-level generation accuracy.

Table 3: Comparisons of High-level generation evaluation on the AMASS and MotionX-W test sets.

| Method | AMASS | | | | | MotionX-W | | | | |
|---|---|---|---|---|---|---|---|---|---|---|
| | FID $\downarrow$ | R@1 $\uparrow$ | R@3 $\uparrow$ | MMDist $\downarrow$ | DIV $\rightarrow$ | FID $\downarrow$ | R@1 $\uparrow$ | R@3 $\uparrow$ | MMDist $\downarrow$ | DIV $\rightarrow$ |
| **Base Model** | 0.29 | 0.47 | 0.78 | 3.06 | 10.59 | 0.21 | 0.39 | 0.75 | 2.60 | 10.53 |
| **SFT-Filter** | 0.25 | 0.48 | 0.79 | 3.02 | 10.62 | 0.20 | 0.40 | 0.75 | 2.58 | 10.55 |
| **RobotMDM** | 0.28 | 0.47 | 0.78 | 3.08 | 10.45 | 0.22 | 0.39 | 0.74 | 2.60 | 10.55 |
| **GT** | – | 0.52 | 0.80 | 2.96 | 10.37 | – | 0.46 | 0.81 | 2.23 | 10.14 |
| **RLPF** | 0.29 | 0.47 | 0.76 | 3.10 | 10.57 | 0.24 | 0.39 | 0.74 | 2.64 | 10.45 |

## 4.4 ABLATION ANALYSIS

We conduct ablation studies to analyze the contribution of RLFS's key components.

To answer **Q2 (Reward Analysis)**, we compare two variants: **RLPF-Full** (our complete method), and **RLPF-w/o track** (ablated by removing the motion tracking reward). As shown in Tables 4, 5, and 6, the experimental results demonstrate that RLPF-Full achieves higher success rates than RLPF-w/o-track across both the IsaacGym and MuJoCo simulators on the AMASS and MotionX-W benchmarks, indicating that the motion tracking reward is crucial for enhancing the physical feasibility of the generated motions.

To answer **Q3 (Alignment Verification)**, we further compare RLPF-Full with another variant **RLPF-w/o align**, which removes the alignment verification across both AMASS and MotionX-W benchmarks. The results in Table 4, 5 and 6 show that alignment verification is crucial for maintaining motion generation accuracy. It's also important to note that RLPF-w/o align exhibits near-complete loss of semantic correspondence, as evidenced by the visualizations in Figure 3.

Table 4: Ablation results of high-level generation on the AMASS and MotionX-W test sets.

| Method | AMASS | | | | | MotionX-W | | | | |
|---|---|---|---|---|---|---|---|---|---|---|
| | FID ↓ | R@1 ↑ | R@3 ↑ | MMDist ↓ | DIV → | FID ↓ | R@1 ↑ | R@3 ↑ | MMDist ↓ | DIV → |
| **RLPF-Full** | 0.29 | 0.47 | 0.76 | 3.10 | 10.57 | 0.24 | 0.39 | 0.74 | 2.64 | 10.45 |
| **RLPF-w/o track** | 0.23 | 0.49 | 0.79 | 3.02 | 10.64 | 0.20 | 0.42 | 0.77 | 2.54 | 10.64 |
| **RLPF-w/o align** | 41.97 | 0.07 | 0.15 | 7.57 | 8.53 | 28.74 | 0.08 | 0.15 | 7.46 | 8.91 |

Table 5: Ablation results of low-level tracking on IsaacGym and MuJoCo simulators (AMASS Dataset).

| Method | IsaacGym | | | MuJoCo | | |
|---|---|---|---|---|---|---|
| | $Succ \uparrow$ | $E_{mpjpe} \downarrow$ | $E_{mpkpe} \downarrow$ | $Succ \uparrow$ | $E_{mpjpe} \downarrow$ | $E_{mpkpe} \downarrow$ |
| **RLPF-Full** | 0.83 | 0.21 | 0.08 | 0.78 | 0.25 | 0.26 |
| **RLPF-w/o track** | 0.50 | 0.31 | 0.21 | 0.46 | 0.44 | 0.58 |
| **RLPF-w/o align** | 0.99 | 0.18 | 0.07 | 0.99 | 0.22 | 0.13 |

Table 6: Ablation results of low-level tracking on IsaacGym and MuJoCo simulators (MotionX-W Dataset).

| Method | IsaacGym | | | MuJoCo | | |
|---|---|---|---|---|---|---|
| | $Succ \uparrow$ | $E_{mpjpe} \downarrow$ | $E_{mpkpe} \downarrow$ | $Succ \uparrow$ | $E_{mpjpe} \downarrow$ | $E_{mpkpe} \downarrow$ |
| **RLPF-Full** | 0.75 | 0.26 | 0.11 | 0.71 | 0.35 | 0.19 |
| **RLPF-w/o track** | 0.47 | 0.32 | 0.18 | 0.42 | 0.47 | 1.10 |
| **RLPF-w/o align** | 0.95 | 0.18 | 0.08 | 0.93 | 0.26 | 0.14 |

## 5 CONCLUSION

In this paper, we present Reinforcement Learning from Physical Feedback (RLPF), a novel framework that resolves physical feasibility in motion generation models for humanoid robots. RLPF integrates two key components, a pre-trained motion tracking policy and an alignment verification module, which ensures that generated motions are both physically feasible and semantically aligned with the input instructions. Extensive experiments demonstrate the effectiveness of our RLPF approach, showing a pathway for the humanoid robots in real-world applications.

**Limitations and Future Work.** The current framework's generalization is constrained by its frozen motion tracking policy, which is pre-trained on a limited static dataset. A promising direction for future research would be to jointly train the large motion generation model with an adaptable tracking policy, potentially enabling robust handling of out-of-distribution motions while maintaining physical feasibility.

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

# A    VISUALIZATIONS OF RLPF-W/O ALIGN

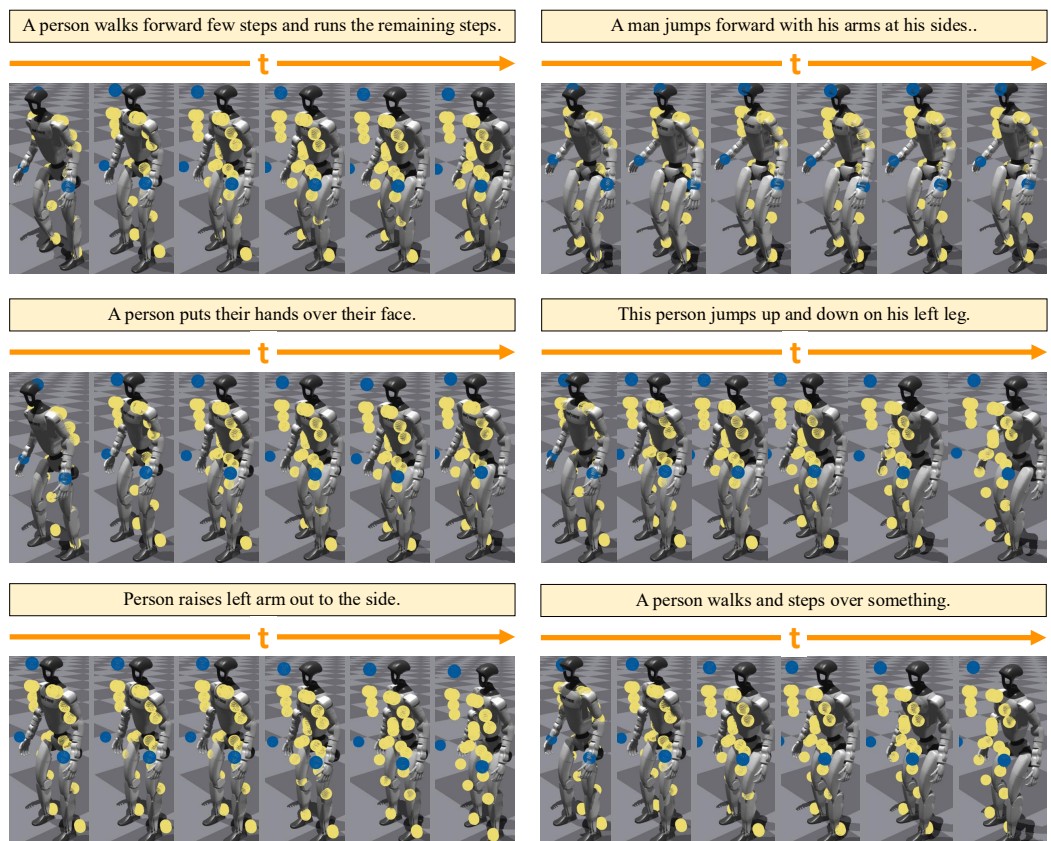

Figure 3: Visualizations of RLPF-w/o align. Since training relies solely on the motion tracking reward, the model predominantly generates standing motion sequences, losing semantic alignment with the input textual instructions.

# B    ADDITIONAL RESULTS

## B.1    ADDITIONAL VIDEO RESULTS

The results of more comprehensive and clear comparisons, particularly real-world videos, presented in accordance with anonymity requirements, are available at https://anonymous.4open.science/r/RLPF/.

## B.2    VISUALIZATIONS OF RLPF-W/O ALIGN

As discussed in line 298 of the main text, although **RLPF-w/o align** performs very well in low-level evaluations, it nearly completely loses semantic correspondence. In high-level generation evaluations, all metrics exhibit very poor performance. Visualizations are illustrated in Figure 3.

# C    EVALUATION DETAILS

## C.1    SIMULATOR DETAILS

We use the Isaac Gym and MuJoCo simulators to conduct our experiments. The two simulation environments are illustrated in Fig. 4.

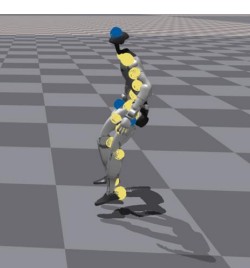 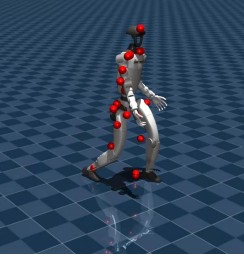 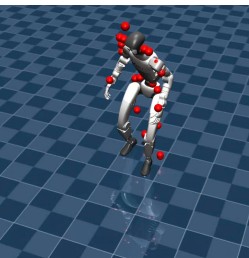

(a) Isaac Gym example 1. (b) Isaac Gym example 2. (c) MuJoCo example 1. (d) MuJoCo example 2.

Figure 4: Example images of the Isaac Gym and MuJoCo simulators.

### C.1.1 ISAAC GYM

Isaac Gym (Makoviychuk et al., 2021) is a high-performance physics simulation environment developed by NVIDIA, designed for robotic reinforcement learning (RL) tasks. It uses GPU acceleration to implement an end-to-end training process, significantly improving the simulation and training speed of complex robotic tasks. Isaac Gym supports importing standard robot model formats (such as URDF and MJCF), provides Tensor APIs for direct GPU interaction, and can simulate multiple environment instances in parallel, which is critical for reinforcement learning.

### C.1.2 MUJOCO

MuJoCo (Multi-Joint dynamics with Contact) (Todorov et al., 2012) is an open source physics engine designed for simulating complex physical systems involving multi-joint structures and contact interactions. It is widely used in the fields of robotics, biomechanics, and machine learning (especially reinforcement learning). MuJoCo is known for its unique combination of speed, accuracy, and modeling capabilities, and is particularly suitable for model-based optimization and contact-rich simulation scenarios.

### C.1.3 SIMULATOR TRANSFER

As shown in Humanoid-Gym (Gu et al., 2024), the dynamics in MuJoCo are closer to the real environment than Isaac Gym. Therefore, following previous work on motion tracking policies (Ji et al., 2024; He et al., 2025), we first conduct large-scale reinforcement learning training in IsaacGym. To evaluate the robustness and generalization of the policy, we then perform a zero-shot transfer to the MuJoCo simulator to verify its transferability. Finally, we deploy the policy and generated motions on a humanoid robot in the real world to verify the effectiveness of RLPF for the motion tracking task.

## C.2 EVALUATION METRICS DETAILS

### C.2.1 LOW-LEVEL TRACKING METRICS

For the motion tracking process, we follow OmniH2O (Ji et al., 2024) and PHC (Luo et al., 2023), and adopt the following metrics: Success Rate, MPJPE, and MPKPE.

- **Success Rate**, denoted as $Succ$, measures whether the humanoid robot can follow the reference motion without losing balance. Tracking is deemed unsuccessful if the average deviation from the reference trajectory exceeds 0.5 meters at any time point, or if the root pitch angle surpasses a specified threshold.

- **Mean Per Joint Position Error (MPJPE)**, denoted as $E_{mpjpe}$ (rad), evaluates the accuracy of joint tracking. It quantifies the average error in degrees of freedom (DoF) rotations between the reference motion and the tracking trajectory.

- **Mean Per Keybody Position Error (MPKPE)**, denoted as $E_{mpkpe}$ (m), evaluates the accuracy of keypoint position tracking. It quantifies the average positional error between keypoints in the reference motion and those in the tracking trajectory.

### C.2.2 HIGH-LEVEL GENERATION METRICS

Following Guo et al. (2022), we adopt text-motion retrieval metrics (R@1, R@2, R@3), Multimodal Distance (MMDist), and Fréchet Inception Distance (FID).

- **Text-Motion Retrieval Metrics (R@1, R@2, R@3)**. These metrics assess the quality of text-to-motion generation by evaluating how accurately the generated motions correspond to input text instructions in a retrieval setting. R@1 (Recall at 1) represents the proportion of cases in which the correct motion is ranked as the top match for a given text query. It reflects the model's capability to retrieve the most relevant motion. R@2 and R@3 follow the same principle, indicating the frequency with which the correct motion appears within the top 2 and top 3 retrieved results, respectively.

- **Multimodal Distance (MMDist)**. MMDist measures the average distance between the feature representations of the generated motion and the corresponding text instruction in a shared embedding space. Typically, embeddings are obtained using pretrained text-motion-retrieval models, and a distance metric is then applied. A lower MMDist indicates better alignment between the text and motion, suggesting that the generated motion closely matches the semantic content of the text.

- **Fréchet Inception Distance (FID)**. FID is a widely used metric to evaluate the quality of generated motion by comparing the distribution of generated motions to that of ground-truth motions. It computes the Fréchet distance between two multivariate Gaussian distributions modeled on the feature representations of ground-truth and generated motions, typically obtained using a pretrained motion feature extractor (e.g., an Inception-like network). A lower FID score signifies that the generated motions more closely resemble real motions in both quality and distribution, indicating higher realism and fidelity.

## D   TRAINING DETAILS

### D.1   DATASET DETAILS

To enhance the text-to-motion generation capabilities of the LLM backbone, we pre-trained it on the large-scale MotionX dataset (Lin et al., 2023), and the motion tokenizer was trained on the same data. MotionX includes data from high-quality optical motion capture and relatively lower-quality motion estimation models derived from third-person videos.

During RL fine-tuning, to further enhance the physical feasibility of the generated motions, we used fully motion-captured datasets, CMU (Carnegie Mellon University, 2007) and AMASS (Mahmood et al., 2019), which are subsets of MotionX. The quantities of the datasets are summarized in Table 7.

Table 7: Comparison of Dataset Quantities

| Dataset | Motion Sequence |
|---------|-----------------|
| MotionX | 81,082 |
| AMASS   | 13,145 |
| CMU     | 5,458 |

### D.2   MOTION RETARGETING DETAILS

As described in line 154 of the main text, we follow the idea of H2O (He et al., 2024) and adopt a two-step optimization-based approach to achieve the retargeting process.

Since SMPL parameters represent various human body shapes, we first optimize the shape parameter $\beta'$ to approximate a humanoid form. We select 14 body links corresponding between humans and humanoids, as shown in Table 8, and perform gradient descent on $\beta'$ to minimize joint distances in the rest pose.

Using the optimized shape $\beta'$ along with the original translation $p$ and pose $\theta$, we perform gradient descent to further reduce the distances between corresponding body links. This process ensures accurate motion retargeting and generates robot motion sequences.

Table 8: Correspondence between Humanoid and Human Body Links

| Humanoid Links Name | Human Body Links Name |
|---|---|
| Pelvis | Pelvis |
| Left hip pitch link | Left hip |
| Left knee link | Left knee |
| Left ankle roll link | Left ankle |
| Right hip pitch link | Right hip |
| Right knee link | Right knee |
| Right ankle roll link | Right ankle |
| Left shoulder roll link | Left shoulder |
| Left elbow link | Left elbow |
| Left hand link | Left hand |
| Right shoulder roll link | Right shoulder |
| Right elbow link | Right elbow |
| Right hand link | Right hand |
| Head link | Head |

### D.3 HYPER-PARAMETER SETTING AND EXPERIMENTAL COMPUTATIONAL RESOURCES

The specific parameter settings in the experiment are shown in Table 9. For computational resources, we use eight A800 GPUs to conduct our experiments.

Table 9: Hyperparameters of Large Motion Models

| Hyperparameters | Value |
|---|---|
| Model Version | NousResearch/Llama-2-7b-hf |
| Per Device Train Batch Size | 1 |
| Gradient Accumulation Steps | 1 |
| Num Generations | 20 |
| Max Prompt Length | 100 |
| Max Completion Length | 100 |
| Max Grad Norm | 0.1 |
| Reward Weight Tracking | 10 |
| Reward Weight Align | 2 |
| Weight KL | 1.0 |
| KL Type | k3 |

### D.4 EMBODIMENT DETAILS

In this work, we use Unitree G1 humanoid robot to conduct experiments. The robot is equipped with an onboard Jetson Orin NX, which serves as the primary computing and communication unit. The control policy takes motion tracking targets as input, computes the desired joint positions for each motor, and transmits commands to the robot's low-level interface. The control policy runs at an inference frequency of 50 Hz. The low-level interface operates at 200 Hz, ensuring smooth real-time control. Communication between the control policy and the low-level interface is implemented using LCM (Lightweight Communications and Marshalling) (Huang et al., 2010).

## D.5 LLM Usage

Our use of LLMs in this paper includes two parts: writing assistance and dataset construction. During writing, we use LLMs to help polish the expression of some texts in our paper. For data construction process, we use LLMs to choose the walking-relevant motion-text pairs.

