# OpenReview forum: "RL from Physical Feedback: Aligning Large Motion Models with Humanoid Control"
_ICLR.cc/2026/Conference — ICLR 2026 Conference Withdrawn Submission_

### Official Review · Reviewer_4vQn · 2025-10-27

**Soundness:** 2
**Presentation:** 4
**Contribution:** 2
**Rating:** 4
**Confidence:** 4

**Summary:**

This work addresses the problem of learning large text-to-motion (T2M) models that achieve both semantic fidelity and physical feasibility. The main contribution is Reinforcement Learning from Physical Feedback (RLPF), a framework that fine-tunes T2M models using reinforcement learning with two complementary rewards: a physics-based reward derived from a robot policy’s tracking performance, and a semantic alignment reward computed from the encoded representations of the input text and the generated motion. Experiments conducted on the AMASS and MotionX-W datasets, evaluated in IsaacGym, MuJoCo, (and real-world deployment on the Unitree G1 humanoid,) show that RLPF achieves significantly higher success rates and lower joint/keypoint errors than all baselines, while maintaining competitive semantic alignment metrics.

**Strengths:**

- The paper is well-structured and clearly written, making it easy to follow and understand.
- It addresses an important and timely problem at the intersection of large motion models and humanoid robot control.
- The proposed RLPF framework is novel, introducing the integration of reinforcement learning–based physical feedback into text-to-motion generation, effectively bridging the gap between semantic motion synthesis and physically executable control.

**Weaknesses:**

- The overall soundness of the proposed method is not entirely convincing. The definition of physical feasibility heavily depends on whether an RL-trained tracking controller can follow the generated and retargeted motion. This design raises several conceptual concerns.

- It is unclear whether the teacher–student framework is necessary. If the teacher policy can successfully track a reference trajectory but the student cannot due to missing privileged information, should the motion still be considered physically feasible?

- The notion of feasibility appears to be controller-dependent. For instance, if a general-purpose tracking controller fails to track a motion but a specialized policy trained for that single trajectory can succeed, should the trajectory be deemed feasible? This dependence makes the physical feedback evaluation ambiguous.

- The feasibility score may also depend on robot-specific parameters, such as actuator gains or control frequency. Adjusting these parameters could make more trajectories trackable, suggesting that the reward signal might not reflect intrinsic motion feasibility but rather the robot’s hardware configuration.

- The necessity of using a real humanoid robot for physical feedback is also questionable. Since the feedback is derived entirely from simulation, a simulated humanoid character could have been used instead, avoiding the complexity of motion retargeting.

- In the experimental section, while the paper claims to include hardware experiments, it would be more convincing to provide comparative videos demonstrating real-world tracking performance versus baseline methods.

**Questions:**

Please check the weakness part.

---

### Official Review · Reviewer_pcaX · 2025-10-31

**Soundness:** 3
**Presentation:** 2
**Contribution:** 2
**Rating:** 2
**Confidence:** 3

**Summary:**

This paper proposes a method called Reinforcement Learning from Physical Feedback (RLPF), which finetunes a pre-trained text-to-motion model using reinforcement learning (RL) with rewards derived from a motion tracker. To ensure that generated motions align with the input text descriptions, RLPF introduces a secondary module termed the alignment verification module. The paper demonstrates the method’s contribution through experiments on the AMASS and MotionX datasets.

**Strengths:**

* This work addresses an important and timely problem. Current text-to-motion models can generate high-quality kinematic motions consistent with text prompts. However, these motions are often not directly transferable to physical hardware due to issues such as limited physical feasibility and challenges in retargeting. Developing algorithms that produce robust, high-quality motions that can be readily transferred to physical robots is therefore a valuable research direction.
* Writing quality for the introduction and the method section is good. Text is easy to follow and the problem being studied is well motivated.

**Weaknesses:**

* The propsed method seems over complicated to me. The idea of using RL to fine-tune a text-to-motion model based on some physical feasibility/transferability reward makes sense. But some of the later choices for the method are not well justified. For example: 1) Why is the base text-to-motion (T2M) model an LLM when most state-of-the-art (SOTA) T2M models are diffusion-based? 2) Why would a reward model derived from a tracking policy ensure that kinematic motions are better suited for physical systems? (It may make sense in simulation but not necessarily in real-world settings.) 3) Finally, the main approach does not appear to produce strong results (as seen in the ablation results in Table 5), prompting the addition of an alignment module to keep generated motions closer to the text descriptions.

* Following on the above point, I am curious how this method compares with recent motion tracking and humanoid control works such as BeyondMimic. BeyondMimic focuses on transferring kinematic motions to real hardware. Given that modern T2M models are already capable of generating high-quality kinematic motions, a robust motion tracker might suffice to bridge text to physical motion. Considering the higher quality of motions generated by these recent methods, the novelty and contribution of this work appear limited.

* The humanoid motions shown on the anonymous webpage also appear to be of low quality. The robot exhibits considerable jittering and instability compared to the source kinematic motions. Moreover, when compared to recent methods such as BeyondMimic, the results look significantly worse.

* In addition to weak qualitative results, the empirical evidence also does not convincingly demonstrate the method’s contribution. For example, Table 3 shows little to no improvement over the baselines. The experimental section is also difficult to follow, with several missing details and limited discussion of the obtained results.

**Questions:**

1. The output robot motions (videos) appear unnatural, with substantial jittering, shaking, and foot sliding. Can the raw kinematic motions be transferred to the robot using a standard retargeting method? If yes, showing those as a comparison could help highlight the contribution of the proposed method.

2. The paper repeatedly refers to “large motion models.” Is this an established term? If it derives from the use of an LLM backbone, this should be clearly explained early in the paper.

3. Figure 2 requires improvement. The symbols used in the figure are not explained in the caption, forcing the reader to search for their meaning throughout the text, where they are inconsistently described. This makes the overview figure difficult to interpret. Some terms mentioned in the figure are also missing from the text, which adds confusion.

4. A more basic question: assuming a standard VQ-VAE setup, is the codebook trained jointly with the encoder and decoder? If so, how did you choose the number of codes and initialize them?

5. Why did you choose a motion tokenizer and an LLM for the T2M model when most SOTA approaches use diffusion models?

6. Why do Tables 1 and 2 refer to IsaacGym and MuJoCo as the test tasks, while the paper describes these results as coming from AMASS and MotionX? What do these tables actually represent?

7. Can you provide insight into why results in Tables 1 and 2 are better than those in Table 3?

8. Why does the alignment module have such a large impact, especially on a metric like FID? Including videos of the ablation results would be very helpful to show how the absence of the alignment module affects motion quality—does it primarily reduce consistency with the text prompt, or does it substantially degrade overall motion quality?

**Details Of Ethics Concerns:**

.

---

### Official Review · Reviewer_R9UM · 2025-11-01

**Soundness:** 2
**Presentation:** 2
**Contribution:** 2
**Rating:** 2
**Confidence:** 4

**Summary:**

This paper proposes Reinforcement Learning from Physical Feedback (RLPF), a novel framework that fine-tunes large text-to-motion models using reinforcement learning guided by physics-based feedback. The method introduces two main components: (1) a motion tracking policy that evaluates the physical feasibility of generated motions, and (2) an alignment verification module to preserve semantic consistency between text and motion. The paper demonstrates significant improvements in simulation-based metrics for both AMASS and MotionX datasets, with limited real-world deployment results on the Unitree G1 humanoid robot.

**Strengths:**

- Clear motivation and formulation: The paper addresses a meaningful gap between text-driven motion generation and physically executable humanoid control, a problem of growing importance for humanoid generalization.

- Novel learning framework: The use of reinforcement learning from physics feedback (as an analogy to RLHF) is conceptually strong and well-motivated.

**Weaknesses:**

- Lack of qualitative baselines: The paper does not include visual qualitative comparisons in simulation or real-world deployment against baselines. Visual demonstrations would better support the claimed improvements in physical realism.

- Limited real-world evaluation: Only four real-world results are presented, which is quite limited for a robotics paper. The sample size is too small to validate generalization or reliability outside simulation.

- Dependency on weak tracking policy: The improvements may largely stem from using a relatively weak base tracking policy. State-of-the-art humanoid motion trackers such as GMT, TWIST, or CLONE already achieve near-perfect motion imitation; with those as baselines, the proposed RLPF may show negligible benefit. An ablation on the base tracking policy should be added to prove or refute this.

- Missing discussion on language-conditioned control: The related work omits discussion of language-conditioned humanoid control papers such as LangWBC[1], UH-1[2], and LeVERB[3], which are directly relevant to aligning high-level instructions with low-level humanoid behavior.

- Unclear scalability and efficiency: The paper does not specify how many RLPF iterations or total training time were required. This makes it hard to judge the computational cost and practical feasibility of the proposed pipeline.

- Bounded by retargeting and frozen tracking: Since the physical feedback is mediated by a pre-trained tracking policy and motion retargeting, the final performance remains limited by these components’ quality and domain coverage.

[1] LangWBC: Language-directed Humanoid Whole-Body Control via End-to-end Learning

[2] Learning from Massive Human Videos for Universal Humanoid Pose Control

[3] LeVERB: Humanoid Whole-Body Control with Latent Vision-Language Instruction

**Questions:**

- How many RLPF fine-tuning iterations and total training hours are needed for convergence?

- How would RLPF perform if the base tracking policy were replaced with a state-of-the-art one (e.g., GMT, TWIST, CLONE)?

- Could the authors provide visual qualitative comparisons (videos or snapshots) against baselines to better illustrate physical feasibility?

---

### Official Review · Reviewer_FFzc · 2025-11-01

**Soundness:** 3
**Presentation:** 3
**Contribution:** 2
**Rating:** 4
**Confidence:** 5

**Summary:**

This paper proposes Reinforcement Learning from Physical Feedback (RLPF), a framework to adapt large text-to-motion models for humanoid execution by integrating (i) a motion-tracking policy as a physics-based reward and (ii) a semantic alignment module to prevent collapse toward trivial motions. The idea is to use simulation feedback to fine-tune a pre-trained motion generator, improving physical feasibility while preserving text-motion intent.

**Strengths:**

- Addresses a real gap between text-driven motion generation and humanoid-feasible control. This is an open problem, and the integration of physics feedback into generative motion models is a meaningful exploration.
- Technical pipeline is clean and modular — motion tokenizer + LLM + motion-tracking RL + contrastive alignment.
- Sim2real real robot experiments on Unitree G1 support some of the claims.

**Weaknesses:**

- The method largely reuses known components—VQ-motion tokenizer, supervised T2M pretraining, teacher–student tracking policies, GRPO fine-tuning—and combines them. The RL formulation mirrors RLHF but uses physics reward instead of human preference. Conceptually sensible but not that novel.
- Results do not convincingly justify the added complexity. While the paper reports improvements in feasibility, the absolute success rates remain moderate (e.g., IsaacGym 0.83 for AMASS, 0.75 for MotionX-W) and real-world demos appear limited qualitatively** ￼. The benefit relative to filtered SFT baselines is present but not dramatic.
    - While the resulting human motion is of good quality, the humanoid motion quality is quite low, with many small steps and instability.

**Questions:**

How sensitive is the method to the specific motion-tracking policy quality? If the policy improves or changes, does the generative model need to be RL-tuned again?

---

### Note · Authors · 2025-11-13

I have read and agree with the venue's withdrawal policy on behalf of myself and my co-authors.